# Acute Stress Effects over Time on Gene Expression Patterns in the Male Zebrafish (*Danio rerio*) Brain

**DOI:** 10.3390/ani15162431

**Published:** 2025-08-19

**Authors:** Constanze Pietsch, Jonathan Konrad, Paulina Pawlak

**Affiliations:** 1School of Agricultural, Forest and Food Sciences (HAFL), Bern University of Applied Sciences (BFH), 3052 Zollikofen, Switzerland; jonathankonrad@gmx.ch (J.K.); paulina.pawlak@bfh.ch (P.P.); 2ProFishCare GmbH, Bösch 43, 6331 Hünenberg, Switzerland

**Keywords:** animal welfare, abiotic stress, stress regulation, marker genes, handling

## Abstract

The present study presents gene expression patterns in differently stressed male zebrafish. These investigations are important for our understanding of stress responses in a fish species that is frequently used in research. The investigation of the stress responses over time represents unique insights into the effects of exposure to acute stress over time. Gene expression analyses have been conducted to reveal genes contributing to the outcome of stress application the most. The results show that genes of different pathways strongly contribute to the stress responses in male zebrafish.

## 1. Introduction

Distress is a common concern in aquaculture and research settings, since the well-being of fish is influenced [1,2,3]. Fish are regularly exposed to different stressors that include short-lived, acute stressors, such as handling, and chronic stressors that affect the fish over longer periods of time, e.g., compromised water quality. In addition to negative stressors, evidence is accumulating that also routine events such as feedings may induce stress responses in fish [4,5]. Furthermore, several studies indicate that stress responses at the brain level of fish show species-specific differences [6,7,8,9]. For zebrafish, it has been observed that the hunger level also influences the level of aggressiveness in females but not in males [10], which indicates that gender-specific differences exist in stress responses that also affect their behavior. Hence, investigating the responses to different stressors only in males first is an important contribution to our current knowledge on stress responses in fish, especially if the different regulation levels at the fish brain level are focused on.

One of the main pathways involved in stress responses of fish is the hypothalamic-pituitary–interrenal (HPI) axis [11]. It regulates the secretion of cortisol, the primary stress hormone in fish, which is involved in the coordination of physiological responses to stressors and allows the fish to cope with stress [12]. Activation of the HPI axis begins with the release of corticotropin-releasing factor (Crf), leading to the synthesis of proopiomelanocortin (Pomc) and stimulation of the interrenal cells to produce cortisol [13,14]. However, the availability of Crf and urotensin (Uro 1), which is also a ligand for the Crf receptors [15,16], is strongly regulated by corticotropin-releasing hormone-binding protein (Crh-bp, [17]). In addition, the effects on the *crh-bp* expression in the fish brain have been found to be species-specific and also depend on the quality and duration of stress [7,18,19]. Nevertheless, both distress due to handling and feeding-related practices have been shown to induce stress responses in fish. These may include anticipatory behaviors, which include increased locomotor activity at varying time points before the feeding event [20]. But increased aggression between individuals during and after feeding has also been observed [21,22]. As mentioned above, zebrafish may show increased aggressiveness during feeding, which may be influenced by the hunger level, especially in females [10].

Generally, elevated plasma cortisol levels are highly conserved responses to stress across vertebrates and are widely used as a marker to assess the intensity of the stress responses [23]. Therefore, stress hormone measurements are also included in the present investigation for comparison. However, gene expression patterns in the fish brain may add important information about the contribution of different regulation pathways to the overall stress responses. In addition to the HPI axis, neural activity in the brain can also be assessed as changes in the expression of immediate early genes (IEGs). These genes typically have low expression levels during resting states but quickly respond to neuronal activation [24]. Furthermore, stress commonly impacts the expression of genes related to appetite regulation and feeding behavior in fish [25,26]. Crf appears to be one of the central factors for the anorectic effects in fish, and its location in the brain overlaps with the regions responsible for feed intake [27,28]. Its mRNA levels increase after hypoxia or handling [29,30], but the effects of feeding events or feed omission on *crf* expression have less frequently been investigated in fish [31]. Hence, the present investigation represents an important contribution to our understanding of the HPI axis activation in different stress scenarios. Furthermore, propiomelanocortin (Pomc) and cocaine- and amphetamine-regulated transcript (Cart) are strong anorexigenic factors in the fish brain [32,33,34]. Interestingly, intraperitoneal cortisol treatment resulted in a dose-dependent up-regulation of the mRNA levels of the gene encoding for neuropeptide Y (Npy) in the forebrain of goldfish, *Carassius auratus* [35], thereby allowing influences on appetite regulation in fish. Furthermore, the mRNA levels of both, *npy* and ghrelin (*ghrel*), another orexigenic factor in fish, increased in the zebrafish brain upon exposure to acute stress [36]. Together, these orexigenic and anorexigenic factors allow the fine-tuning of the appetite and feeding behavior. In addition, neuropeptides such as galanin (Gal) have also been shown to be involved in the regulation of stress responses in the zebrafish hypothalamus [37]. By investigating the expression levels of different orexigenic and anorexigenic factors, the present study adds further knowledge on interactions of the stress axis and appetite regulation pathways in differently stressed zebrafish. 

Moreover, serotonergic, dopaminergic, and opioid pathways are also involved in stress responses of fish [38,39,40]. Similarly, involvement of the isotonergic system in the regulation of the responses to different stressors was shown in several fish species [41,42,43]. Additionally, the opioid system mediates stress-induced analgesia, which subsequently may further influence stress regulation in fish [9,44]. However, the involvement of opioid receptors in stress responses of different brain parts in fish has not yet been monitored over time.

There are considerable differences in the stress responses of male and female zebrafish [9], indicating that the same set of genes is not relevant for the development of stress responses in female and male zebrafish. To further elucidate this, the present study was conducted only with males. However, the aim of this study was also to investigate the dynamics of gene expression regulation in four brain parts of adult male zebrafish after exposure to acute stress to allow comparison with other fish species, as well as with female zebrafish in the future. The effects of stress are known to be dynamic responses that are different for stressors of different quality and intensity. Hence, the current identification of genes that specifically respond to certain stress situations within 30 to 90 min after application of a stressor profoundly improves our understanding of stress responses in fish. 

## 2. Materials and Methods

### 2.1. Rearing Conditions

Prior to the experiment, the zebrafish (*Danio rerio*) belonging to wild-type x AB strain F1 generations were reared in a 180 L glass tank (Juwel Rio 180, purchased from Hornbach Baumarkt AG, Sursee, Switzerland) equipped with two biofilters (Eheim Aussenfilter Professionel 4+ 250 and a Juwel Pumpe Eccoflow 1500, both purchased from Hornbach Baumarkt AG, Sursee, Switzerland). The fish were up to 12 months old at the start of the experiment. All fish were fed with fresh *Artemia* nauplii every day (Ocean Nutrition Europe, Essen, Belgium) in the morning between 8:00 and 8:30 a.m. and in the afternoon with commercial dry feeds (Zebrafeed, 200–400 mm size, by Sparos, Olhão, Portugal). From these fish, three males for each treatment in duplicate were transferred to 2.3 L rearing tanks (MC1B, Zebcare by Fleuren & Nooijen, Nederweert, The Netherlands) connected to a recirculating aquaria system including a moving bed biofilter. The results from the regular monitoring of water temperature, oxygen levels, pH, and conductivity levels can be found in Section A.1 and Section A.2. The fish were reared under these conditions for 3 days, with the feeding regime described above continued.

### 2.2. Experimental Design

After acclimatization as described above, the control fish were taken directly from the 2.3 L tanks before the scheduled feeding. The remaining fish were either exposed to air by netting for 1 min (air), with euthanasia started 30, 60, or 90 min after that, or fed in the morning (feed), left unfed but receiving the filtered water in which the *Artemia* nauplii had been thawed (feed contr), chased with a net for 1 min (chas), or confined with a net for 1 min (conf). These treatments had been selected because distress and a positive stimulus such as feed-rewarding can lead to different stress regulations in fish, as has already been shown for common carp [45]. For the current study, all fish were euthanized with an overdose of tricaine methanesulfonate (>150 mg/L MS-222, Sigma-Aldrich, Buchs, Switzerland). From all fish, the weight (Mettler Toledo GmbH, Greifensee, Switzerland), the total length, and the standard body length were recorded. From these values the Fulton’s condition factors were calculated by dividing the weight of each fish by its total length cubed × 100 [46]. Furthermore, body mass indices (bmi) were calculated (weight [g]/standard length [mm]^2^) for each fish. The heads of the fish were stored in RNAlater^®^ (Sigma-Aldrich, Buchs, Switzerland) for at least 24 h before the brains were cut into four different parts (telencephalon, hypothalamus, optic tectum, and rhombencephalon) by using a stereomicroscope (VWR^®^ VisiScope^®^ STB150, VWR International, Dietikon, Switzerland). All experiments and procedures had previously been approved by the Local Animal Care Committee under license no. BE69/2020 and were in accordance with the guidelines set by the Swiss Council on Animal Care.

### 2.3. qPCR Analyses

The primer pairs that were used for the gene expression studies can be found in Section B.1 and had already been used in earlier studies [8]. All amplicons had a size ranging from 110 bp to 200 bp. Prior to all analyses, the primer pairs were validated, and the respective PCR products were confirmed by Sanger sequencing [9] supported by the Microsynth AG (Balgach, Switzerland). For this, a cleanup of the PCR products using NucleoSpin Gel and PCR Clean-up kit (Macherey-Nagel AG, Urdorf, Switzerland) was performed. The sequencing results were compared to known sequences via the basic local alignment search tool (Blast-N) function from the National Center for Biotechnology Information (NCBI, https://www.ncbi.nlm.nih.gov/ accessed on 2 May 2024). The primer efficiencies were checked by LinRegPCR (version 2020.2; available at https://medischebiologie.nl/files/, accessed on 24 February 2024) and are reported in Section B.1.

RNeasy Micro Kits (Qiagen AG, Hombrechtikon, Switzerland) were used for total RNA extraction from each of the four brain parts, followed by determining the RNA content on a plate reader (Tecan Infinite M200 Pro, Tecan Instruments, Crailsheim, Germany). After that, 2 μL of total RNA were reverse-transcribed using the ProtoScript ReverseTranscription Kit (NEB, distributed by BioConcept AG, Allschwil, Switzerland) and pre-amplified with the PreAmp Master Mix from Fluidigm Corporation (South San Francisco CA, USA) as described earlier [8]. After that, the pre-amplified cDNA was diluted four-fold, mixed with the SsoFast™EvaGreen^®^ Supermix with Low ROX master mix (Bio-Rad Laboratories AG, Cressier, Switzerland) and the Fluidigm sample reagent (Fluidigm Corporation, South San Francisco, CA, USA) according to the manufacturer’s protocol, loaded on BioMark™ Dynamic Array™ integrated fluidic circuit (IFC) plates (192 × 24, Fluidigm Corporation, South San Francisco, CA, USA), and incubated in a Juno^TM^ system (Fluidigm Corporation, South San Francisco CA, USA) for 33 min. Subsequently, real-time qPCRs were run on a BioMark^TM^ system (Fluidigm Corporation, South San Francisco CA, USA). The cycling conditions included an initial activation at 95 °C for 10 min followed by 30 two-step cycles (denaturation at 95 °C for 10 s and annealing/extension at 60 °C for 1 min).

The initial results for each PCR run were calculated with Fluidigm Real-Time PCR Analysis software (version 4.8.1, Fluidigm Corporation). This was followed by determination of the optimal reference genes for each brain part based on a set of 7–12 potential reference genes using RefFinder (https://www.ciidirsinaloa.com.mx/RefFinder-master/, accessed on 18 June 2025); more details are given in Section C.1. Subsequently, the individual gene expression values were calculated relative to the expression of the reference genes. This was followed by calculation of the normalized fold-change in expression of each target gene relative to the control group as described by Taylor et al. [47].

### 2.4. Analysis of Body Corticosteroid Levels

Individual body homogenates, except for the heads of the fish, were prepared manually with a handheld homogenizer (D1000, Merck & Cie, Buchs, Switzerland) and were extracted twice with diethylether. Furthermore, the extracts were purified on C18 columns (SEP-PAK VAC C18 1CC 100/BX, Waters AG, Dättwil, Switzerland), and cortisol and cortisone were analyzed using a UHPLC–MS/MS system [31]. The individual steroid values were calculated as ng steroid per g body weight.

### 2.5. Statistical Analysis

The gene expression values derived from each brain part were used to build mixed models with a fully Bayesian approach using the brms package [48] in R studio (Version 1.2.1335) for each stressor separately, as explained earlier by Burren & Pietsch [49]. The results from the models included point estimators and the individual SEMs, as well as the posterior predictive *p* values. The significance was determined using Wald χ2-statistics for the generalized linear models and F-statistics for the mixed models. Finally, the estimated marginal means and the 95% credible intervals were calculated. A *p* < 0.05 was considered statistically significant. Figure 1, Figure 2, Figure 3 and Figure 4 show only significant gene expression differences from these calculations. Additionally, principal component analyses (PCA) were performed to identify certain gene clusters that typically correspond to the respective treatments. All PCA were performed in R studio on individual gene sets as described earlier [31]. For a given variable, the sum of the cos2 on all principal components is equal to one. For the present study, the contribution of the two main components to the total variance as displayed by the cos2 values is given for the individual treatment groups.

## 3. Results

### 3.1. Morphological Differences Between the Fish and Whole-Body Steroid Levels

As expected, the morphology of the adult zebrafish that were used for the experiment showed no significant difference between the treatment groups (Section D.1 and Section D.2). Hence, an effect of the treatment group assignment was not observed. The analysis of the body stress hormone levels revealed that the levels of 11-deoxycorticosterone and corticosterone in the whole-body samples were not quantifiable. Changes of the cortisone and cortisol levels in the body homogenates were higher in fish belonging to the chasing, confinement, and air-exposure groups than in fish that had been feed-rewarded or fish from the feed-control group (Section E.1). The two-sided ANOVA showed no significant effect of treatment on the cortisol levels. However, there was a significant influence of the timepoint of sampling after application of the stressors on the steroid levels (*p* = 0.024; Section E.2).

### 3.2. Immediate Early Genes and Metabolic Genes

The immediate early genes in the telencephalon showed different gene expression patterns for all acute stressors that had been used (Figure 1). Only the distressed fish and the feed-control fish displayed a mutual down-regulation of *c-fos* 60 min after treatment. Furthermore, the feed-control fish were the only ones showing an up-regulation of *citrsyn, egr1, erk1*, and *gapdh*. Moreover, the gapdh expression was not changed in the other brain parts that were investigated. In addition, chased fish showed an up-regulation of *palld*. The feed-control fish showed a down-regulation of this gene in the telencephalon, and also, in the remaining brain parts, only down-regulation of *palld* was observed. The detailed expression pattern visualizations for the telencephalon can be found in Appendix A, uploaded in the repository. 

In the hypothalamus, only feed-reward, feed-control, and chased fish showed decreases of *palld* (Figure 1). The expression of *egr1* displayed a difference between chased and feed-control fish, with a decreased in the chased fish and an up-regulation of this gene in the feed-control fish. In addition, a down-regulation *c-fos* was absent in the feed and the air-exposure groups compared with the remaining treatment groups. A detailed visualization of expression pattern in the hypothalamus can be found in Appendix A, uploaded in the repository.

**Figure 1 animals-15-02431-f001:**
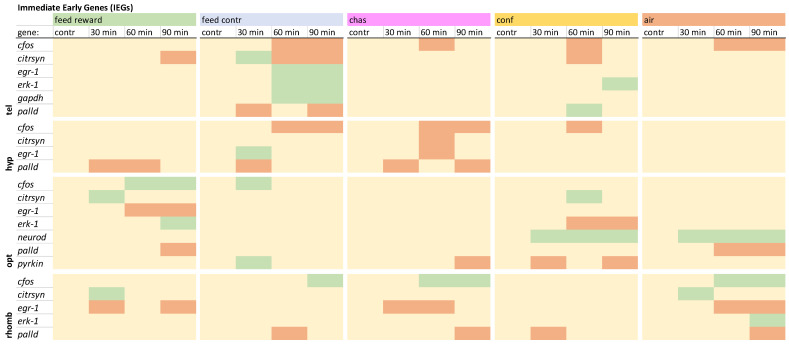
Overview of the expression profiles of early immediate genes (IEGs) in the telencephalon (tel), hypothalamus (hyp), optic tectum (opt), and rhombencephalon (rhomb) of male zebrafish 0, 30, 60, and 90 min after the treatment, whereby feed-rewarding, feed-control treatment, chasing, confinement, and air exposure were used as stressors. N = 6 per treatment. The significance value *p* < 0.05 was calculated from the Bayes models performed on the log2 values of the normalized gene expression values; yellow = not changed, red = down-regulated, green = up-regulated.

In contrast to those in the telencephalon and the hypothalamus, the gene expression patterns of *c-fos* in the optic tectum were increased in the feed and the feed-control groups (Figure 1). The expression of *citrsyn* was only higher in the feed-reward and confinement groups. The expression of *egr-1* was only down-regulated in the feed-rewarded fish 60 and 90 min after treatment. In addition, *erk-1* was only up-regulated in feed-rewarded fish and down-regulated in confined fish. Furthermore, *neurod* was up-regulated in all chased and air-exposed fish. In addition, *pyrkin* showed up-regulation in the feed-control fish but down-regulation in chased and confined animals. The detailed expression pattern visualizations can be found in Appendix A, uploaded in the repository.

The IEGs in the rhombencephalon showed increased expression of *c-fos* 60 min after chasing and air exposure, as well as 90 min after treatment in the feed-control, chasing, and air-exposure groups (Figure 1). The expression of *citrsyn* was only higher in the feed-reward and air-exposure groups. The expression of *egr-1* was not down-regulated in the feed-control or the confinement treatment. In addition, *erk-1* was only up-regulated in air-exposed fish 90 min after treatment. Finally, *palld* did not show a down-regulation in the feed-rewarded fish compared with the remaining treatments. The detailed expression pattern visualizations can be found in Appendix A, uploaded in the repository.

### 3.3. HPI Axis-Related Genes

The HPI axis-related genes in the telencephalon showed a down-regulation of *crf2* in the feed-reward, confinement, and air-exposure groups, whereas the expression of the *crf-r2* was down-regulated at different time points in all treatments (Figure 2). Furthermore, the expression of *gr* was only changed in the feed-control group. Furthermore, *pomc A* was only up-regulated 30 min after chasing, whereas pomc B was up-regulated 60 and 90 min after feed-control treatment. The expression of *uro 1* was only down-regulated in the feed-control and confinement groups. The detailed visualization of the expression patterns in the telencephalon can be found in Appendix A, uploaded in the repository. 

**Figure 2 animals-15-02431-f002:**
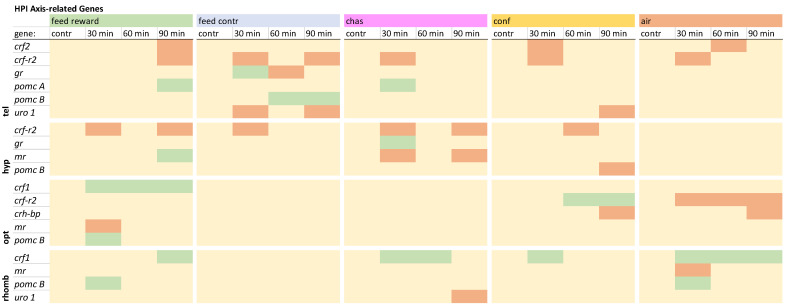
Overview of the gene expression profiles of HPI axis-related genes in the telencephalon (tel), hypothalamus (hyp), optic tectum (opt), and rhombencephalon (rhomb) of male zebrafish 0, 30, 60, and 90 min after the treatment, whereby feed-rewarding, feed-control treatment, chasing, confinement, and air exposure were used as stressors. N = 6 per treatment. The significance value *p* < 0.05 was calculated from the Bayes models performed on the log2 values of the normalized gene expression values; yellow = not changed, red = down-regulated, green = up-regulated.

In the hypothalamus, the expression of *crf2* was not changed, but all groups showed down-regulation of *crf-r2* except for the air-exposed fish (Figure 2). An up-regulation of *gr* was only noted for chased fish 30 min after treatment. An up-regulation of *mr* was observed in feed-rewarded fish 90 min after treatment, whereas chased fish showed down-regulation of this gene. In addition, *pomc B* was only down-regulated in confined fish 90 min after treatment. A detailed description of the expression pattern in the hypothalamus can be found in Appendix A, uploaded in the repository.

The investigation of the gene expression patterns of the HPI axis-related genes in the optic tectum revealed that *crf1* was only up-regulated in all feed-rewarded fish (Figure 2). Furthermore, the expression of *crf-r2* was down-regulated in this brain part in air-exposed fish but up-regulated in confined fish, while these two groups both showed a down-regulation of *crh-bp*. While *gr* changes were observed in the telencephalon and hypothalamus, the optic tectum showed only a down-regulation of *mr* in the feed-rewarded fish and a *pomc B* up-regulation in the optic tectum. The detailed expression pattern visualization for this brain part can be found in Appendix A, uploaded in the repository.

The evaluation of HPI axis-related genes in the rhombencephalon revealed that the feed-control fish were the only treatment group that did not show an up-regulation of *crf1* in this brain part (Figure 2). Changes of the *gr* were also absent in the rhombencephalon, and the expression of *mr* was only down-regulated 30 min after air exposure. The expression of *pomc B* was up-regulated in the feed-reward and air-exposure groups 30 min after treatment. Furthermore, the expression of *uro 1* was only down-regulated in chased fish 90 min after treatment. The detailed expression pattern visualization can be found in Appendix A, uploaded in the repository.

### 3.4. Appetite-Related Genes

The investigation of appetite-related genes in the telencephalon revealed that *agouti* was not down-regulated only in the feed-rewarded fish (Figure 3). These fish were also the only ones that showed down-regulation of *cart* and up-regulation of *cck-b* 90 min after treatment. The expression of *grp* was only down-regulated in chased fish 60 min after treatment. Furthermore, an up-regualtion of *npy* was observed in feed-rewarded and chased fish, whereas confined fish showed a down-regulation of this gene 90 min after treatment. The detailed expression pattern visualizations can be found in Appendix A, uploaded in the repository.

**Figure 3 animals-15-02431-f003:**
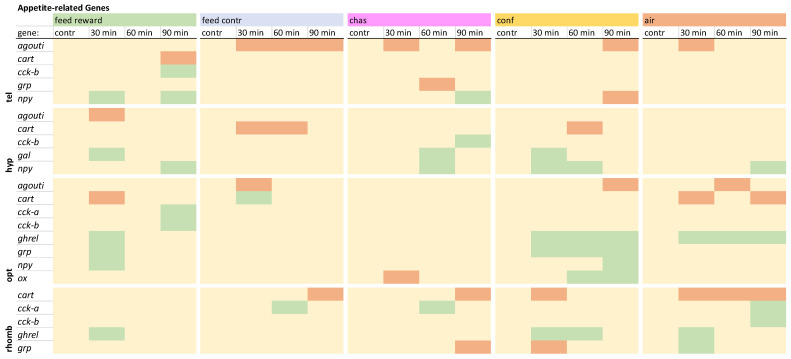
Overview of the gene expression profiles of appetite-related genes in the telencephalon (tel), hypothalamus (hyp), optic tectum (opt), and rhombencephalon (rhomb) of male zebrafish 0, 30, 60, and 90 min after the treatment, whereby feed-rewarding, feed-control treatment, chasing, confinement, and air exposure were used as stressors. N = 6 per treatment. The significance value *p* < 0.05 was calculated from the Bayes models performed on the log2 values of the normalized gene expression values; yellow = not changed, red = down-regulated, green = up-regulated.

In the hypothalamus, *agouti* was only down-regulated in feed-rewarded fish 30 min after treatment (Figure 3). The expression of *cart* was only down-regulated in some of the feed-control and confinement groups. Only chased fish showed an up-regulation of *cck-b* 90 min after treatment. An up-regulation of *gal* was absent in the feed-control and the air-exposure groups in this brain part. The expression of *grp* was not changed in any of the treatments, but the expression of *npy* was increased in all treatments at different time points except for the feed-control fish. The detailed expression pattern visualizations can be found in Appendix A, uploaded in the repository.

The investigation of the gene expression patterns of the appetite-related genes in the optic tectum revealed that the expression of *agouti* was not down-regulated in the feed-reward and chased groups (Figure 3). The expression of *cart* was down-regulated in feed-rewarded and air-exposed fish but up-regulated in feed-control fish 30 min after treatment. *Cck-a* and *-b* were only up-regulated in feed-rewarded fish 90 min after treatment. *Ghrel* and *grp* showed similar expression patterns in feed-rewarded and confined fish. In addition, the expression of *npy* was only increased in these two treatment groups. Finally, the expression of *ox* was found to be changed only in the optic tectum, where it was increased in confined fish and decreased in chased animals. The detailed visualization and description of the expression patterns for this brain part can be found in Appendix A, uploaded in the repository.

The appetite-related genes in the rhombencephalon showed no signficant changes of *agouti,* although changes of this gene have been observed in the other three brain parts. In addition, no down-regulation of *cart* was observed in the feed-rewarded fish, but was noted in all remaining treatment groups (Figure 3). The expression of *cck-a* was up-regulated in the feed-control fish and chased fish 60 min after treatment and in the air-exposed fish 90 min after treatment, whereas the expression of *cck-b* was only increased in air-exposed fish. The expression of *ghrel* was not increased in the feed-control fish and the chased fish. In addition, the expression of *grp* was down-regulated in chased and confined fish at different time points and up-regulated in air-exposed fish 30 min after treatment. The detailed expression pattern visualizations can be found in Appendix A, uploaded in the repository.

### 3.5. The Gene Expression Patterns of the Serotonergic and Dopaminergic Genes, Opioid, Isotocin, and Prolactin Receptors

The remaining genes that were investigated in the telencephalon showed that *iso pre* was not increased in the feed-rewarded fish (Figure 4), whereas the *iso-r1* showed down-regulation only in the feed-rewarded, the feed-control, and the confined fish at different time points. The expression of *mtor* was only decreased in feed-control fish 30 min after treatment. The *opio d1b* expression was not decreased in the feed-rewarded fish or the air-exposed animals. The *prola* showed increased expression in the feed-rewarded fish 90 min after treatment and down-regulation in feed-control and confined fish 60 and/or 90 min after treatment. Finally, the *tph* expression was down-regulated in all treatments, but at different time points. The detailed expression pattern visualizations can be found in Appendix A, uploaded in the repository.

**Figure 4 animals-15-02431-f004:**
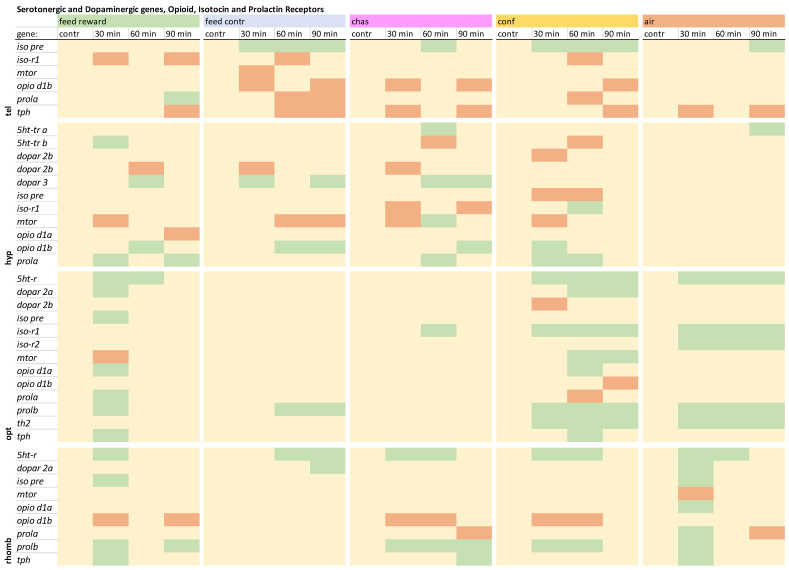
Overview of the gene expression profiles of the serotonergic and dopaminergic genes, opioid, isotocin and the prolactin receptors in the telencephalon (tel), hypothalamus (hyp), optic tectum (opt), and rhombencephalon (rhomb) of male zebrafish 0, 30, 60, and 90 min after the treatment, whereby feed-rewarding, feed-control treatment, chasing, confinement, and air exposure were used as stressors. N = 6 per treatment. The significance value *p* < 0.05 was calculated from the Bayes models performed on the log2 values of the normalized gene expression values; yellow = not changed, red = down-regulated, green = up-regulated.

In contrast to the telencephalon, the expression of *5ht-tr a* and *b* was changed in the hypothalamus, displaying increased *5ht-tr a* expression in chased and air-exposed fish and increased *5ht-tr b* expression in the feed-reward group (Figure 4). The expression of *5ht-tr b* was decreased only in the chased and confined fish 60 min after treatment. Changes of the dopamine receptors have also not been observed in the telencephalon, but in the hypothalamus, confined fish showed a decreased expression of *dopar 2a* 60 min after treatment. In addition, the expression of *dopar 2b* was decreased in feed-reward, feed-control, and chased fish 30 and/or 60 min after treatment. The expression of *dopar 3* was increased in these treatment group at different time points, but not in confined or air-exposed fish. Furthermore, the expression of *iso pre* was only decreased in confined fish, which also showed an increased *iso-r1* expression 60 min after treatment. The expression of *mtor* was down-regulated in all treatments at different time points except for air-exposed fish. In addition, only chased fish showed an increased *mtor* expression 60 min after treatment. The *opio d1a* expression was only decreased in the hypothalamus of the feed-rewarded fish, whereas the expression of *opio d1b* was found to be increased in feed-rewarded, chased, and confined fish at different time points after treatment. Finally, the *prola* expression was not increased in the feed-control and air-exposure groups but was increased in the remaining treatment groups at different time point after treatment. The detailed expression pattern visualizations can be found in Appendix A, uploaded in the repository.

The gene expression patterns of the remaining genes in the optic tectum showed increased *5ht-r* expression in feed-rewarded, confined, and air-exposed fish (Figure 4). The expression of *dopar 2a* was increased in feed-rewarded and confined fish, but the expression of *dopar 2b* only down-regulated in confined fish 30 min after treatment. *Iso pre* was only up-regulated in feed-rewarded fish 30 min after treatment. The expression of the *iso-r1* was only up-regulated in distressed fish, and only all air-exposed fish showed an up-regulation of *iso-r2*. In addition, the expression of *mtor* was down-regulated in feed-rewarded fish 30 min after treatment and up-regulated in confined fish 60 and 90 min after treatment. Furthermore, the expression of *opio d1a* was found to be up-regulated in feed-rewarded and confined fish, whereas the expression of *opio d1b* was down-regulated only in confined fish 90 min after treatment.

An increased *prola* expression was observed in feed-reward fish 30 min after treatment and a down-regulation of this gene in confined fish 60 min after treatment. In contrast, *prolb* was up-regulated in this brain part in all treatments groups at different time points after treatment except for the chased fish. Changes of the *th2* expression were only observed in the optic tectum, and only the confined and air-exposed fish showed increases of its gene expression. In contrast to the telencephalon, which showed down-regulation of *tph*, the optic tectum displayed up-regulation of this gene in the feed-rewarded and confined fish. The detailed expression pattern visualizations and descriptions can be found in Appendix A, uploaded in the repository.

The expression of *5ht-r* in the rhombencephalon showed increased values in all treatments but at different time points after treatment (Figure 4). The expression of *dopar 2a* was increased only in the feed-control and air-exposure groups. Furthermore, the expression of *iso pre* was increased in the feed-reward and air-exposure groups. The expression of *mtor* and *opio d1a* was changed only in air-exposed fish 30 min after treatment. In contrast, the expression of *opio d1b* was down-regulated in feed-rewarded, chased, and confined fish at different time points after treatment. The expression of *prola* was down-regulated only in chased fish 90 min after treatment and up-regulated in air-exposed animals 30 min after treatment. In contrast, all treatment groups showed up-regulation of *prolb* in the rhombencephalon at different time points after treatment except for the feed-control fish. Finally, up-regulation of *tph* was not observed in the feed-control and confinement groups. The detailed expression pattern visualizations can be found in Appendix A, uploaded in the repository.

### 3.6. Principal Component Analyses Revealing Gene Expression Patterns

The principal component analyses (PCA) for all genes and brain parts have been uploaded in the repository. Based on these results, the two most important genes for each brain regulation pathway were selected and used for additional PCA. For the genes in the telencephalon, *uro 1* and *mtor* were identified as genes with a strong relevance for the expression patterns in this brain part (Figure 5). In contrast, the contribution of *pomc B* to the outcome of gene expressions in the telencephalon was considerably low, and genes such as *cck-a* and *cck-b* repeatedly are among the most contributing genes in this brain part. 

Moreover, the genes with the highest influence in the hypothalamus of the male zebrafish were appetite-related genes, such as *agouti* and *gal*, but the different stressors resulted in different genes contributing to the stress responses in this brain part (Figure 6). 

In the optic tectum of males, *crh-bp* often had the strongest ability to explain the variability in the data sets obtained from the analysis of gene expression patterns after application of the different stressors (Figure 7). The other genes with strong contributions to the outcome in this brain part varied for each stressor. 

Finally, the rhombencephalon showed the strongest influence of the genes *ckap-5* and *succdh* (Figure 8). 

Selecting the optimal set of genes from these analyses for each brain part and each treatment separately resulted in an overview of genes that mostly contributed to the gene expression patterns in Figure 9. This figure revealed that a mixture of genes contributed to the stress responses the most. The most frequently mentioned IEGs across all treatments and all brain parts were *succdh, gapdh,* and *egr-1*. Similarly, the most frequently mentioned HPI axis-related genes were *uro-1* and *crh-bp*. The appetite genes with the highest frequency were *cck-a* and *-b* and *npy*. From the remaining regulatory pathways in the brain, only *gaba ra* was mentioned as frequently as the genes belonging to the IEGs, the HPI axis, or the appetite-regulating genes. For future investigations of gene expression patterns in the brain of stressed zebrafish, it is, therefore, recommended to use the most frequently contributing genes mentioned here to allow an optimal assessment of the early stress responses in this species.

## 4. Discussion

In this study, acute negative stressors and feed reward were used to examine the different effects of stressors on male zebrafish brains. There were no significant differences in the measured morphological parameters between the males from the different treatment groups (Section D.1 and Section D.2). However, the fish that were distressed appeared to respond with higher stress hormone levels in the body than fish belonging to the feed group or the feed-control group. Obviously, the quality of the stressor determines if there is a considerable stress hormone response in zebrafish. That air exposure but not feed-rewarding resulted in a significant cortisol response has also been observed for common carp (*Cyprinus carpio*) 30 min after treatment [31]. Hence, the cortisol and cortisone responses to stress are not only dependent on the quality of the stressor, although the quality of the stressors in combination with the intensity of the treatments probably was not optimal to show more differences in steroid levels between the different treatment groups in the present study. The study of Ramsay et al. [50] also indicated that feeding affects the cortisol levels in whole-body homogenates of crowded adult zebrafish, which may indicate that similar effects may also have occurred in the present study, using zebrafish that have not been fed for one day before sampling, except for the feed-reward group. But the present study clearly showed that stress hormone levels are time-dependent, as has been observed when investigating the influence of the timepoint of sampling after application of the stressors on steroid levels. A time-dependent effect of stressors on cortisol levels of zebrafish has already been described [6,51]. This makes it difficult to choose the optimal time point for stress hormone level analyses in experimental studies. 

In addition, it has been assumed for a long time that stress hormone levels are sufficient indicators of the stress levels in fish. This may be true, for example, if fish are acutely distressed [52,53]. However, the study of Madaro et al. [54] indicated that heavily stressed fish do not show increased blood stress hormone levels, and the study of Opinion et al. [55] even demonstrated a down-regulation of the stress hormone levels in Atlantic salmon, *Salmo salar*, after exposure to long-term stress. Hence, the measurement of stress hormones in fish may not correspond to the actual stress level in fish, and gene-expression markers as earlier indicators of stress may be better. Measuring the early stress responses allows the evaluation of stress levels independently of the identity of the stressor. In this way, different types of stressors or interactions of stressors can be assessed. Furthermore, the focus on early stress responses may allow the application of countermeasures in rearing facilities before obvious and irreversible harm has been done to the fish.

### 4.1. Immediate Early and Metabolic Genes as Stress Indicators

IEGs serve as valuable markers for brain neural activity involved in various processes, including stress responses. The feed-control treatment caused the most changes in IEGs expression in the telencephalon, whereby *egr1, erk-1,* and *gapdh* indicated increased neuronal activity, and *c-fos* and *palld* showed decreased expression. Erk1 and Egr1 are involved in memory processing and recall in fish [56,57]. Compared to that of common carp, *Cyprinus carpio* [58], the expression of *gapdh* was less frequently involved in the stress responses in the male zebrafish in the current study. Responses to stress commonly increase energy expenditures and may also involve increased blood glucose levels in fish [31,59], which can explain effects on the expression of genes involved in the carbohydrate metabolism. 

### 4.2. HPI Axis-Related Genes as Markers for Stress

The activity of the HPI axis in different brain parts was visible across all the treatments. Activation of the HPI axis involves Gr and Mr, which mediate the effects of cortisol on the organism. In contrast to most teleosts, zebrafish only have one Gr isoform [60]. The importance of Mr is also emphasized by the fact that a loss of Gr is not lethal in juvenile zebrafish [61]. However, the rapid locomotor response of zebrafish, as a response to environmental stressors, appears to require HPI axis signaling via Gr but does not involve Mr [62]. Although the stressors in this study did not significantly affect the expression of *gr*, except in the feed-control group in the telencephalon, the expression of *mr* was influenced by feed-rewarding, chasing, and air exposure. This emphasizes that not only Gr but also Mr still have an impact on stress-response regulation. This is also supported by Faught & Vijayan [63], suggesting involvement of Mr in energy management during stress responses. Thus, Mr may be important in prolonged stress response in zebrafish [63,64]. This may explain the observed changes of *mr* expression in the hypothalamus, the optic tectum, and the rhombencephalon of different treatment groups.

Elevated *pomc* expression has been observed previously in acutely stressed zebrafish [52]. In this study, the levels of *pomc A* and *pomc B* expression varied across the treatments and brain parts. Interestingly, the levels of *pomc B* in the rhombencephalon increased 30 min after feed reward but also after air exposure. Pomc is known to be a strong anorectic factor, which could explain its increased transcript level not only after stressful events but also after feeding events in the current study. Similar effects were observed upon feeding of Atlantic salmon (*Salmo salar*), which showed increased expression of two *pomc* isoforms within 3 h after receiving feed [65]. In summary, these investigations show that the responses to stress in fish are often similar, but the timing of the responses appears to be different.

The role of Crf in zebrafish exposed to acute stress is evident [9,66]. Uro 1, a paralogue of Crf, has the ability to bind to Crf receptors r1 and r2 [67]. The Crh binding protein appears to co-occur rather with Crf than with Uro 1 in the brain of adult zebrafish, thereby reducing its availability and limiting the further release of Pomc [28,68]. The importance of *uro 1* expression in the zebrafish brain after exposure to acute stress was emphasized by the PCA results in the present study. Although the expression of *crh-bp* did not show frequent differences between the treatment groups, the PCA revealed that this gene strongly contributed to the outcome of the gene expression patterns in the optic tectum. This further confirms that the effects of stress on the *crh-bp* expression in the fish brain are species-specific and depend on the quality and duration of stress [7,18,19], but also are specific for the four brain parts. Therefore, it is recommended for future investigations to also use brain part-specific investigations of the gene expression patterns after stress exposure.

### 4.3. Changes of Appetite-Related Genes After Exposure to Stress

Stress can affect the food intake of fish through the interaction of Npy and Crf neurons [27,29]. Increased *npy* expression in the hypothalamus of zebrafish upon acute handling stress [36] appear to be similar to the *npy* expression patterns in this study after application of negative stressors, but they were also observed in the telencephalon of fed males. Increased levels of *npy* 1.5 h after feeding have also been observed in the whole brain of Atlantic salmon, although the reasons for this remained unclear [65], as the *npy* expression typically increases before feeding or during fasting and decreases after feeding [69,70].

The expression of *ghrel* was increased in the optic tectum at each time point after confinement and air exposure, as well as 30 min after feeding. Our results, thus, align with findings originating from Cortés et al. [36] that showed that the expression of both *npy* and *ghrel* was increased in the zebrafish brain after exposure to acute stress. Cortés et al. [36] proposed that these two pathways may quickly react to acute stress but are not involved in short-term appetite suppression. In addition, the increase in expression of orexigenic genes may be needed to counteract the anorexigenic effects of stress [36].

Similar to our previous study [9], *cart* showed lower expression after air exposure in the optic tectum and rhombencephalon. Cart is involved in several physiological functions, including appetite regulation and stress [34,71,72]. However, *cart* was also down-regulated in the optic tectum 30 min and in the telencephalon 90 min after receiving a feed reward, maybe due to its two-directional effects on appetite by acting as a short-term satiety factor and a long-term starvation factor [73]. Accordingly, fasting for 3 days resulted in down-regulation of *cart* in the hypothalamus of zebrafish in the study of Nishio et al. [74]. Nevertheless, Cart is also involved in the regulation of the energy balance and plays a role in gustation and feeding behavior, which includes processing of diverse sensory stimuli such as olfactory and visual inputs [75,76,77].

In the hypothalamus, increases of the *gal* expression have also been observed in the current study. Gal appears to have multiple functions in the zebrafish brain, including sleep regulation after increased neuronal activity [78], while absence of Gal can lead to neuronal hyperactivity [79]. In addition, this neuropeptide has recently been shown to be involved in fine-tuning the stress responses in the hypothalamus of zebrafish [37]. Thus, it was not surprising that this gene also strongly contributed to the outcome of the gene regulation patterns in the hypothalamus, as revealed by the PCA.

In summary, the observed effects of the different acute stressors on the expression of appetite genes confirm that acute stress affects the appetite gene regulations in a stressor- and time-dependent way, which already has been noted for common carp in a previous study [8]. 

### 4.4. Serotonergic and Dopaminergic Genes

Frustration as a trigger for aggression is a known phenomenon in vertebrates [41,80]. Vindas et al. [81] observed that Atlantic salmon show higher aggression towards conspecifics when feeding was either omitted or delayed. Furthermore, serotonin and dopamine pathways are known factors influencing aggression in social situations and stress regulation [25,38]. In the current study, males belonging to the feed-control group did not show clear patterns in expression of serotonin- and dopamine-related genes. Interestingly, chasing differently affected both *5ht-ra* and *5htr b* in the hypothalamus after 60 min, suggesting a down-regulation of this circuit. This hypothesis can be further supported by the down-regulation of the *crf-r2* and *mr* in the same brain part 90 min after chasing. Furthermore, a decrease in *tph* expression was observed in the telencephalon after chasing. The telencephalon has been reported to be rich in serotonergic neurons, and Tph can be a specific marker for serotonergic activity [82]. Thus, a reduced *tph* expression may suggest a lower supply of serotonin. Nevertheless, lower expression of the Tph gene was also observed after other treatments in the telencephalon. Taken together, this confirms earlier findings that show a contribution of serotonergic pathways to the stress responses in cyprinids [e.g, 8,26].

Interestingly, no significant differences in the expression of dopamine-related genes were found in the telencephalon, whereas the remaining brain parts showed changes in expression of the dopamine receptors. Increased dopaminergic activity was observed in the hypothalamus of tilapia (*Oreochromis mossambicus*) and rainbow trout (*Oncorhynchus mykiss*) as a result of acute stress [83,84], but also in the telencephalon of trout and common carp [8,83]. Moreover, typical dopaminergic responses to starvation have been observed at the level of the hypothalamus in goldfish, whereby the dopaminergic system is inhibited by fasting [85]. In the same species, stimulation of D1 and D2 receptors reduced feeding [86]. Nevertheless, the D2 and D3 receptors investigated in the current study did not reveal a consistent pattern for the specific stress situations. This further suggests that dopamine receptors are dependent on the stressor type and severity but also on the brain part.

### 4.5. Involvement of Opioid, Isotocin, and Prolactin Receptors in Stress Responses

The isotonergic system in fish, although not fully understood in terms of its exact physiological functions, has been shown to be responsive to various stressors [42,87]. In this study, the isotocin precursor-encoding gene was generally up-regulated regardless of the treatment, except for the hypothalamus 30 and 60 min after confinement, where it was down-regulated. Interestingly, a consistent up-regulation of *iso pre* was observed in the telencephalon of male fish after each treatment, except after the feed-reward treatment. On the contrary, the *iso pre* expression was lower in the feed-reward and feed-control groups 60 min after the treatments in the telencephalon of females [9]. One explanation for this may be the differences in numbers of the vasotocin/isotocin-reactive neurons between the sexes, as observed in medaka, *Oryzias latipes* [88]. Furthermore, it has been indicated that isotocin contributes to the mating success of male zebrafish [89]. Thus, increased activation of the isotocin system can be expected, as males from the current experiment had reared together with females several days before the onset of the experiments. Additionally, isotocin impacts social interactions in male goldfish, *Carassius auratus* [90], and the telencephalon is recognized for its role in social behaviors. Nevertheless, it is important to consider that mature nonapeptide measurements might provide a more accurate reflection of its function compared to the expression profiles of unprocessed precursor mRNA [91].

Opioid and prolactin receptors both participate in stress responses; however, they operate through different mechanisms. Opioids affect neurological stress reactions, whereas prolactin influences physiological processes, including water and ion balance, under stress conditions [92,93]. In this study, the expression of the opioid receptor D1B was decreased in the telencephalon and rhombencephalon after chasing and confinement and in the optic tectum after confinement. This supports the assumption that the opioid D1B receptor type plays a role in stress responses [92], although co-regulation with other pathways may occur given the widespread effects of the opioid system on the organism. Furthermore, it is known that regulating prolactin levels is considered to be important during periods of starvation but also during exercise for providing energy [94]. Our results show that prolactin receptors may not be helpful in clearly distinguishing between the different stressors. The *prola* and *prolb* were generally up-regulated at different time points in all brain parts, except in the telencephalon. 

### 4.6. Unsupervised Learning Methods

An earlier study revealed that different genes had strong influences on the PCA results in male and female zebrafish when fish were sampled 60 min after treatment [9]. In the current study, the contribution of the genes was investigated for each treatment separately to show the contribution of the genes over time. Metabolic genes that played important roles in the stress responses were *succdh* and *gapdh*, which are important for the carbohydrate metabolism [31,59], which may indicate changes in the energy expenditure of the body.

IEGs that contributed strongly over the course of the investigations included *erk-1* and *egr-1*, but these two genes were among the most contributing genes in the telencephalon and hypothalamus of fish from almost all treatment groups. Hence, these genes appear to be important for the dynamics of the stress responses in these two brain parts but are less specific for the distinct stressors in other brain parts.

As expected, a number of HPI-related genes contributed to the dynamics of the stress responses in all treatments. As mentioned above, the contribution of *crh-bp* to the stress dynamics in the optic tectum is present in all treatment groups. Crh-bp is known to prevent further Crf signaling through Crh-r1 and -r2, and is, thus, considered to be an important factor regulating the stress axis in fish [15,17]. According to the study of Doyon et al. [30], increased *crf-bp* expression supports a more rapid return to homeostasis after exposure to stress, which may explain why all treatment groups show a high contribution of *crh-bp* to the gene-regulation dynamics in the optic tectum. In addition, dynamics of the *gr* expression were most relevant in the feed-reward and the feed-control groups but not in the remaining distress groups. Recent research has shown that, at least upon chronic stress, Gr activation reduced feed intake in zebrafish [95]. In contrast, the present study shows for the first time that *gr* dynamics may also be important for gene expression regulation during feeding and feed omission.

Furthermore, *dopa 2a* was among the most contributing gene expressions in the stress dynamics of feed-rewarded fish and confined fish. In addition, a contribution of *th2* was observed only in the optic tectum of fed fish, further supporting involvement of the dopamine pathway in feeding and motivation regulation [38,96]. Another pathway that should gain more attention in the fish brain is the balance of glutamate and gamma-aminobutyric acid (GABA) in stress situations. The study of Li et al. [97] showed that heat stress challenges stimulate the glutamate–glutamine pathway to prevent oxidative damage in rainbow trout. Furthermore, our previous study on common carp revealed that feed omission influenced the glutamate levels in the optic tectum, whereas the expression of *gabaa* was increased in different brain parts and not specifically after application of a distinct stressor [8]. In the present study, the expression of *gabaa* was not significantly changed by the treatments but showed high contribution to the overall gene expression patterns in the optic tectum and in the rhombencephalon, as shown by the PCA. For GABA, it was, furthermore, assumed that it provides negative feedback for the HPI axis [98], which may in part explain the high contribution of *gabaa* in the PCA of the present study.

Finally, a number of genes involved in appetite regulation are among the most important genes contributing highly to the gene expression patterns in the differently treated fish. The dynamics of *npy* were highly important in all treatments, which confirms the findings of a previous study on common carp [8]. In contrast, the cocaine- and amphetamine-regulated transcript (Cart), known as a strong anorexigenic factor in the fish brain [34], was among the five most contributing genes in the air-exposed fish. This further confirms that negative stress induces anorexigenic effects in the hypothalamus of fish. In addition, the expression dynamics of *gal* only contributed to the overall expression patterns in the hypothalamus of fish treated with the feed reward and in chased fish. Although this neuropeptide has been described as a modulator of the stress responses in the hypothalamus of zebrafish [37], it remains unclear why the expression dynamics of this gene have only been most relevant for the feed-rewarded and the chased fish. The present study also indicated a high contribution of *cck-a* and *cck-b* on the outcome of the gene expression changes upon stress. The study of Forsman et al. [99] showed that Cck is a short-term post-prandial satiation factor in trout. Furthermore, fasting is known to affect *cck* expression in the brain and the intestine of fish [100,101,102]. In addition, acute stress has been shown to influence the *cck* expression in common carp [26], which may be one regulatory pathway exerting the anorectic effects of stress. These interactions may also explain the present results for the *cck-a* and *cck-b* expression for male zebrafish.

Taken together, our results confirm that acute stress responses in zebrafish are stressor- and species-specific and show distinct dynamics over time. Future investigations will reveal if the stress responses in female zebrafish differ from the gene expression pattern in acutely stressed males.

## 5. Conclusions

Distress differently regulates gene expression in fish. Generally, the male zebrafish showed more differences between gene expression patterns after the different acute stress treatments compared with carp [8,58]. This strongly confirms that gene regulation at the brain level is species- and stress-specific. In addition, the expression patterns were different depending on the quality of the stressors and the timepoint after stressor application. Future investigations in female zebrafish would be required to draw conclusions on gender-specific characteristics in stress regulation.

## Figures and Tables

**Figure 5 animals-15-02431-f005:**
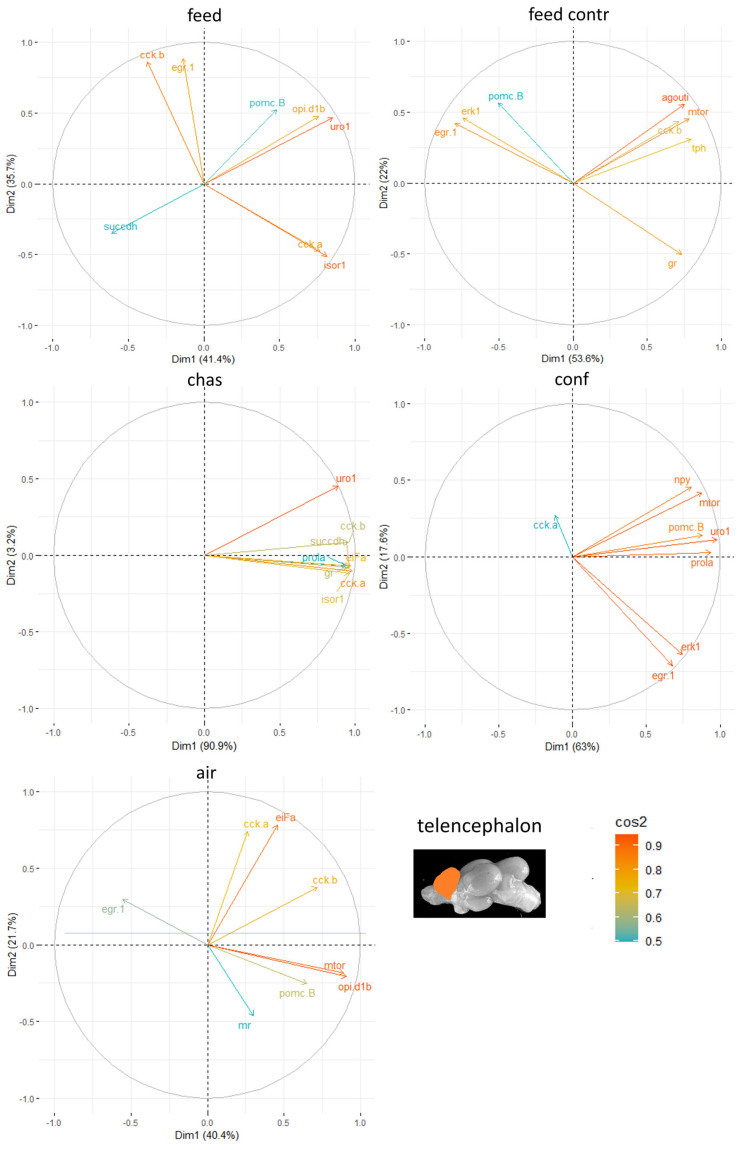
Gene expression analysis with principal component analyses (PCA) for the most contributing genes in each of the telencephalon showing their representation on the factor map as cos2 values, whereby the numbers next to Dim1 and Dim2 indicate the percentage of the variance in the data sets that is explained by the first two components of the PCA of male fish 0, 30 min, 60 min, and 90 min after treatment; n = 6 per treatment.

**Figure 6 animals-15-02431-f006:**
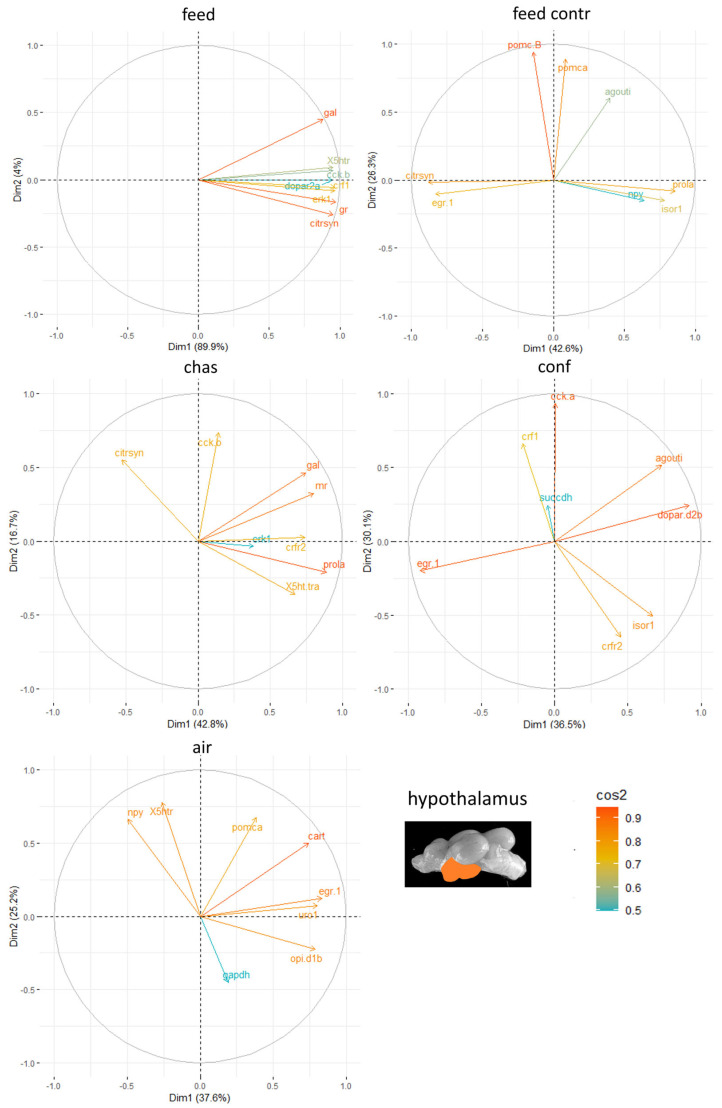
Gene expression analysis with principal component analyses (PCA) for the most contributing genes in each of the hypothalamus showing their representation on the factor map as cos2 values, whereby the numbers next to Dim1 and Dim2 indicate the percentage of the variance in the data sets that is explained by the first two components of the PCA of male fish 0, 30 min, 60 min, and 90 min after treatment; n = 6 per treatment.

**Figure 7 animals-15-02431-f007:**
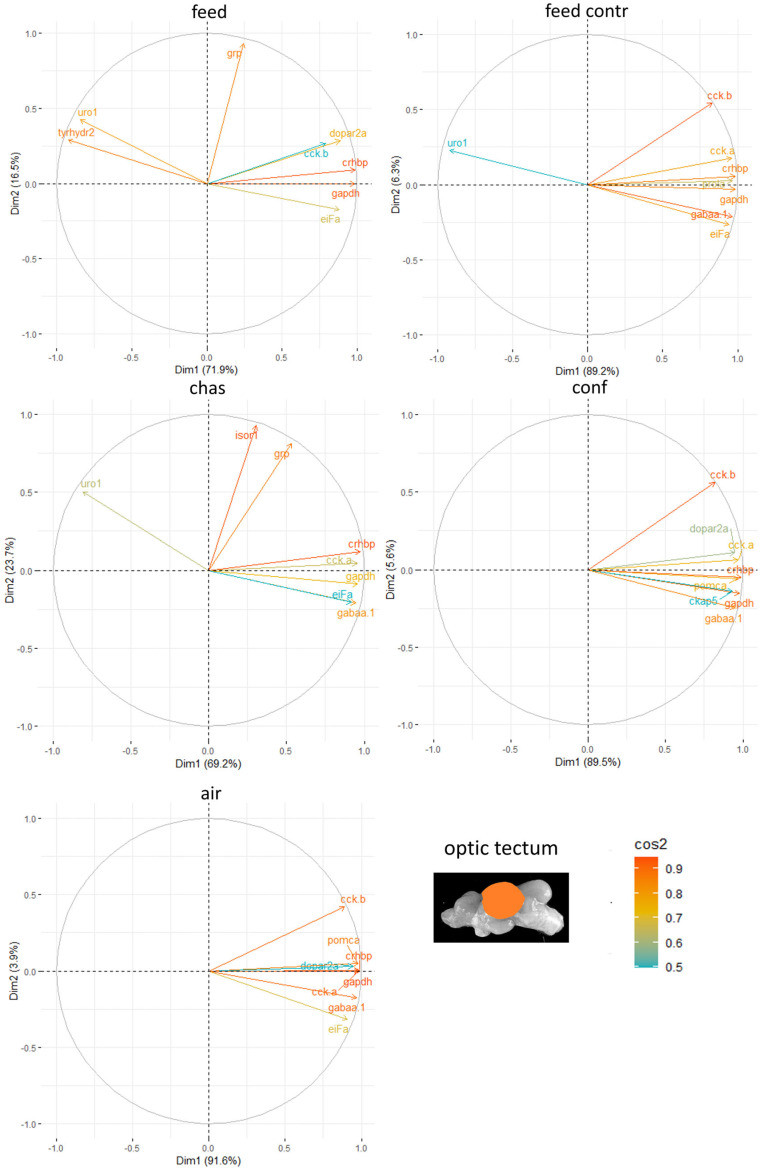
Gene expression analysis with principal component analyses (PCA) for the most contributing genes in each of the optic tectum showing their representation on the factor map as cos2 values, whereby the numbers next to Dim1 and Dim2 indicate the percentage of the variance in the data sets that is explained by the first two components of the PCA of male fish 0, 30 min, 60 min, and 90 min after treatment; n = 6 per treatment.

**Figure 8 animals-15-02431-f008:**
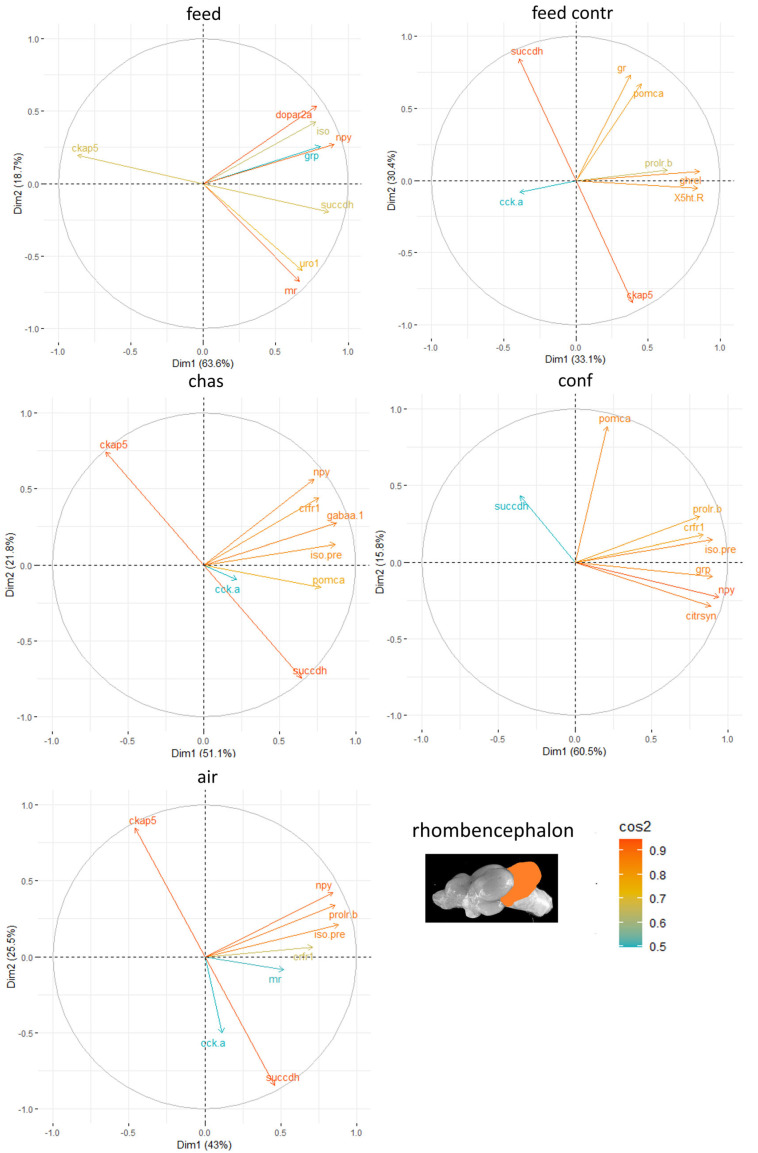
Gene expression analysis with principal component analyses (PCA) for the most contributing genes in each of the rhombencephalon showing their representation on the factor map as cos2 values, whereby the numbers next to Dim1 and Dim2 indicate the percentage of the variance in the data sets that is explained by the first two components of the PCA of male fish 0, 30 min, 60 min, and 90 min after treatment; n = 6 per treatment.

**Figure 9 animals-15-02431-f009:**
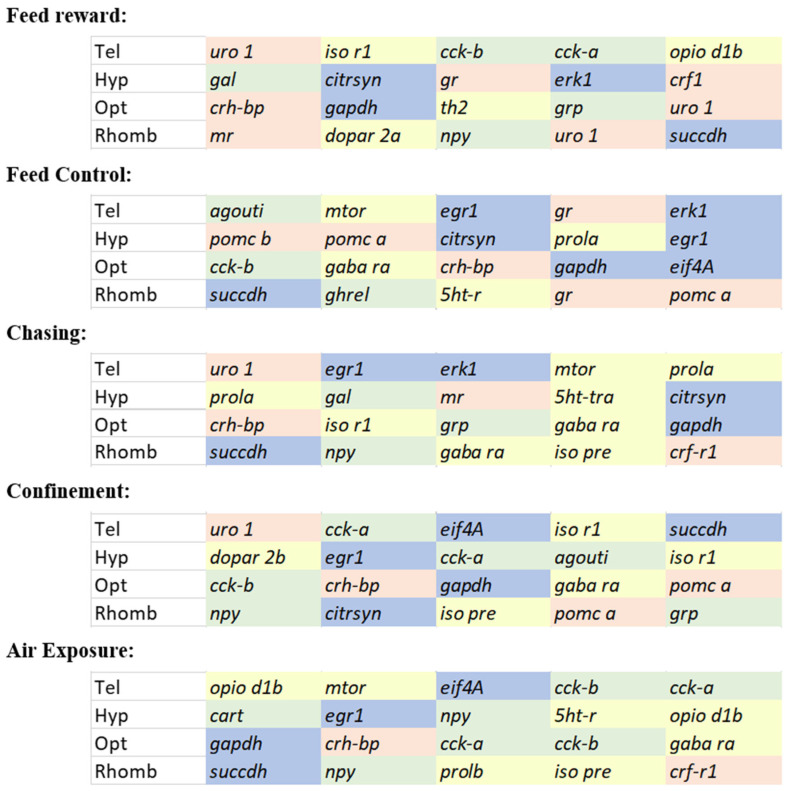
Summary of the 5 most contributing gene to the first two dimensions in the principal component analyses (PCA) in each brain part of male zebrafish 0, 30 min, 60 min, and 90 min after treatment; genes belonging to the group of IEGs and metabolic genes are shaded in blue, HPI axis-related genes are shaded in red, appetite-related genes are shaded in green, and the remaining genes belonging to different pathways, including serotonergic or dopaminergic pathways, or involving isotocin are shaded in yellow.

## Data Availability

All data are available in this repository: https://figshare.com/articles/dataset/Log2_fold_gene_expression_values_and_morphological_details_for_Danio_males/25650624, accessed on 13 August 2025.

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
