# Peer review of "Acute Stress Effects over Time on Gene Expression Patterns in the Male Zebrafish (Danio rerio) Brain"

_animals, 2025, doi:10.3390/ani15162431_

Round 1

Reviewer 1 Report

Comments and Suggestions for Authors

The work presented by Pietsch et al. reveals the effect of exposing male zebrafish to common acute stress factors, evaluating specific changes in gene expression in different regions of the brain. The article is based on a correct methodology, using relevant techniques, and the results obtained are an important contribution to understanding the response to stress in this species.

Bearing in mind the importance normally attributed to cortisol in the assessment of the stress response, how do the authors reconcile the results obtained in cortisol and cortisone levels with the statements in the lines 448-452. Please clarify and detail the interpretation of these results.

In appendix D1 it is important to add the units for a better interpretation of the graphs.

Author Response

The work presented by Pietsch et al. reveals the effect of exposing male zebrafish to common acute stress factors, evaluating specific changes in gene expression in different regions of the brain. The article is based on a correct methodology, using relevant techniques, and the results obtained are an important contribution to understanding the response to stress in this species.

Bearing in mind the importance normally attributed to cortisol in the assessment of the stress response, how do the authors reconcile the results obtained in cortisol and cortisone levels with the statements in the lines 448-452. Please clarify and detail the interpretation of these results.

Answer: Thank you for this comment. The section now makes more clear that the timepoint of investigation is relevant, whereas the treatments were not displaying much difference with respect to the steroid levels. And the following section in the manuscript is also very important, since it cumulates evidence that namely cortisol is not an good indicator of stress in some fish experiments. All changes in the manuscript are highlighted in yellow.

In appendix D1 it is important to add the units for a better interpretation of the graphs

Answer: thank you. The units have been added, but the condition factor is commonly used as a dimensionless index.

Reviewer 2 Report

Comments and Suggestions for Authors

This manuscript described the effects of acute stress on gene expression patterns in male zebrafish brain. In addition, the authors analyzed the correction between several gene groups (IEGs and Appetite, HPI axis-related genes) and stress. These findings could provide valuable insights into potential target genes associated with stress response in fish.

However, there have few weakness:

  1. In introduction, the authors mentioned sex-specific differences in stress responses among zebrafish, with many studies reported females often exhibited more pronounced reactions to stressors like starvation. Why was only male zebrafish selected for this study? This experimental design appears contradictory to the stated research objectives (line 42-43).
  2. The introduction section needed to rewrite, it should focus more precisely on articulating the study's aims, methodological approach, and current advancements in this particular research area.
  3. In methods section, it contained several significant problems regarding experimental description clarity. While numerous references are cited (e.g., for qPCR, HPLC/MS protocols), key methodological details should be described in manuscript.

3.1. Section 2.1, authors should be reorganized with experimental design presented as a separate subsection (e.g., 2.2 Experimental Design).

3.2. Section 2.2 was qPCR, not “PCR” and it required substantial expansion according to MIQE guidelines. At now, many essential information was missing, including: reference gene, primer details (Tm values, product length), calculation method and specific amplification conditions.

  1. The results presentation needed reorganization. Figures should correspond directly with their respective result sections (e.g., results in 3.3 are presented in Figure 5 rather than Figure 1), which creates unnecessary difficulty for readers.
  2. In section 3.1, short-term stressors would not be expected to affect length or weight parameters, so, the morphological analysis was not important. I suggest the authors should add some physiological stress markers, like the expression of blood glucose, lactate levels, and acute-phase inflammatory cytokines (e.g., TNF-α), or oxidative stress indicators (SOD activity, MDA content).
  1. The gene expression analysis across different brain regions lacked methodological justification. HPI-axis related genes were examined inconsistently (crf-2 in telencephalon versus crf-1 in optic tectum). The authors should add the explanation.
  2. Line 141, reference formatting error;

Line 229, "c-fos 60 and/or 90";

Line 377: mtor should be italicized;

All figures need resolution improvement for proper visualization.

Author Response

This manuscript described the effects of acute stress on gene expression patterns in male zebrafish brain. In addition, the authors analyzed the correction between several gene groups (IEGs and Appetite, HPI axis-related genes) and stress. These findings could provide valuable insights into potential target genes associated with stress response in fish.

However, there have few weakness:

  1. In introduction, the authors mentioned sex-specific differences in stress responses among zebrafish, with many studies reported females often exhibited more pronounced reactions to stressors like starvation. Why was only male zebrafish selected for this study? This experimental design appears contradictory to the stated research objectives (line 42-43).

Answer: We have already performed exactly the same investigation with females, but for a publication covering both, males and females, there are too many data. Therefore, the studies have to be published separately with the males being the first ones. The sentence in line 42/43 has, therefore, been changed slightly to emphasize the males in the current manuscript.

Once the study on the females will be published we will highlight all the gender-specific differences in that publication.

  1. The introduction section needed to rewrite, it should focus more precisely on articulating the study's aims, methodological approach, and current advancements in this particular research area.

Answer: Thank you for this comment. The introduction has been revised. We would like to emphasize that it is not an easy task to mention as many different stress regulation levels and influence factors in a short part of the manuscript in a concise format. We have introduced some sentences to make some more connections to the actual study. All changes in the manuscript are highlighted in yellow.

  1. In methods section, it contained several significant problems regarding experimental description clarity. While numerous references are cited (e.g., for qPCR, HPLC/MS protocols), key methodological details should be described in manuscript.

3.1. Section 2.1, authors should be reorganized with experimental design presented as a separate subsection (e.g., 2.2 Experimental Design).

Answer: ok, thank you. This has been done now.

3.2. Section 2.2 was qPCR, not “PCR” and it required substantial expansion according to MIQE guidelines. At now, many essential information was missing, including: reference gene, primer details (Tm values, product length), calculation method and specific amplification conditions.

Answer: All missing details have been added.

  1. The results presentation needed reorganization. Figures should correspond directly with their respective result sections (e.g., results in 3.3 are presented in Figure 5 rather than Figure 1), which creates unnecessary difficulty for readers.

Answer: Ok, the genes have now been sorted as IEGs, HPI axis genes, appetite genes and remaining genes for Fig 1 to 4 including the details for each brain part.

  1. In section 3.1, short-term stressors would not be expected to affect length or weight parameters, so, the morphological analysis was not important. I suggest the authors should add some physiological stress markers, like the expression of blood glucose, lactate levels, and acute-phase inflammatory cytokines (e.g., TNF-α), or oxidative stress indicators (SOD activity, MDA content).

Answer:Well, the morphological data are important to show that there are no differences between the fish with respect to length ect just as a pre-requisite for a valid study. We did not expect stress to affect any of this within some hours. And the fish are very small and clean blood sampling is not an easy task for zebrafish. Moreover, the experimental design already led to a sampling that required a lot of work. Sampling more tissues would have affected the timing of each sample to be taken. So, in summary, it was not possible to do more than is presented.

  1. The gene expression analysis across different brain regions lacked methodological justification. HPI-axis related genes were examined inconsistently (crf-2 in telencephalon versus crf-1 in optic tectum). The authors should add the explanation.

Answer: As mentioned in the Methods section (line 201), only the genes with significant differences are presented in the manuscript. We have measured all genes in all tissues, but not all showed differences between the treatments.

Line 141, reference formatting error;

Answer: Yes, this has been changed now.

Line 229, "c-fos 60 and/or 90";

Answer: This sentence has been clarified now.

Line 377: mtor should be italicized;

Answer: Absolutely, yes.

All figures need resolution improvement for proper visualization.

Answer: We have increased the size and the font size in the figures.

Reviewer 3 Report

Comments and Suggestions for Authors

The manuscript entitled "Acute stress effects over-time on the gene expression patterns in the male zebrafish (Danio rerio) brain" investigates the temporal dynamics of gene expression in different brain regions of male zebrafish following acute stress, which fills a critical gap in understanding sex-specific and brain region-specific stress responses in teleosts. Overall, the study is well-designed with comprehensive analyses covering multiple pathways. However, there are still some problems that need to be explained and modified:

  1. This study was conducted only on male fish, but the introduction section emphasized the significance of gender differences in stress responses. Have similar studies been published on female fish? If not, there should be some data to support why male fish were chosen first. If studies on female fish have already been conducted, the discussion section should focus on discussing the similarities and differences, or at least explore the differences between the unique response patterns of male fish and those of female fish, which will enhance the relevance of the research results.
  2. Lines 127-130 state that the fish brains were divided into 4 regions, and the figure caption also indicates that n=6. Please explain how many biological replicates the article conducted in total, how many fish were used, and how many technical replicates were there for each experiment, because if the sample size n=6 is used for experiments like chasing, confinement, and air exposure, there will still be significant errors and individual differences, and these issues must be clearly and explicitly stated. Increasing the sample size or conducting additional replicates could improve result reliability.  
  3. The font sizes of the coordinate axes in Figures 5 to 8 are too small, and the lines are also not clear.
  4. The manuscriptmentions that there is a high expression of crh-bp in the visual cortex and a bidirectional expression of cart in different brain regions. But there is a lack of detailed mechanism explanations for these phenomena. Further exploration is needed to investigate the connections between these patterns and the known physiological functions.
  5. Lines181-182,the manuscript reports non-significant differences in cortisol levels across treatments, but the association between these hormone levels and the observed gene expression changes remains unclear. A more in-depth analysis of this relationship would enhance the interpretability of results.
  6. The manuscript examines the expression of numerous genes across different brain regions. It would be highly beneficial to provide a concise, conclusive summary of key patterns following each experimental result section. Merely listing the upregulation or downregulation of individual genes suffices to demonstrate transcriptional differences, but the interconnections and relationships among these genes across various stimulus treatments remain unclear. This lack of integrative analysis gives the manuscript a fragmented and unfocused feel. We strongly recommend rationalizing the underlying logic and research framework, and reorganizing the manuscript's structure and writing to enhance coherence and focus.
  7. Appendix A section should be revised to a standard three-line table.

Comments on the Quality of English Language

 The English could be improved to more clearly express the research.

Author Response

The manuscript entitled "Acute stress effects over-time on the gene expression patterns in the male zebrafish (Danio rerio) brain" investigates the temporal dynamics of gene expression in different brain regions of male zebrafish following acute stress, which fills a critical gap in understanding sex-specific and brain region-specific stress responses in teleosts. Overall, the study is well-designed with comprehensive analyses covering multiple pathways. However, there are still some problems that need to be explained and modified:

  1. This study was conducted only on male fish, but the introduction section emphasized the significance of gender differences in stress responses. Have similar studies been published on female fish? If not, there should be some data to support why male fish were chosen first. If studies on female fish have already been conducted, the discussion section should focus on discussing the similarities and differences, or at least explore the differences between the unique response patterns of male fish and those of female fish, which will enhance the relevance of the research results.

Answer: We have already performed exactly the same investigation with females, but for a publication covering both, males and females, there are too many data. Therefore, the studies have to be published separately with the males being the first ones. The sentence in line 42/43 has, therefore, been changed slightly to emphasize the males in the current manuscript. All changes in the manuscript are highlighted in yellow.

Once the study on the females will be published we will highlight all the gender-specific differences in that publication.

  1. Lines 127-130 state that the fish brains were divided into 4 regions, and the figure caption also indicates that n=6. Please explain how many biological replicates the article conducted in total, how many fish were used, and how many technical replicates were there for each experiment, because if the sample size n=6 is used for experiments like chasing, confinement, and air exposure, there will still be significant errors and individual differences, and these issues must be clearly and explicitly stated. Increasing the sample size or conducting additional replicates could improve result reliability.  

Answer: In the section about rearing conditions the relevant details were added. In total, 6 fish per treatment group were analysed. The statistical methods respected the data structure, and errors, for example for the individual gene expression values for each treatment group, are visualized, e.g. in the repository.

  1. The font sizes of the coordinate axes in Figures 5 to 8 are too small, and the lines are also not clear.

Answer: Yes, the Figures have been revised. Thank you for this comment.

  1. The manuscript mentions that there is a high expression of crh-bpin the visual cortex and a bidirectional expression of cart in different brain regions. But there is a lack of detailed mechanism explanations for these phenomena. Further exploration is needed to investigate the connections between these patterns and the known physiological functions.

Answer: additional details about the connection of crf, uro 1 the crf receptors and the crh bp have been added to the discussion.

  1. Lines181-182,the manuscript reports non-significant differences in cortisol levels across treatments, but the association between these hormone levels and the observed gene expression changes remains unclear. A more in-depth analysis of this relationship would enhance the interpretability of results.

Answer: Cortisol was only analysed in whole body homogenates in parallel to the gene expression studies. It would be expected that the analyses of the cortisol levels in blood would have been more conclusive, but blood sampling during the experiments was not possible for this study. In addition, several studies have shown that cortisol levels are not conclusive in stress experiments sometimes. This aspect has already been emphasized in the discussion. Furthermore, we have cited the study of Ramsay et al showing that feeding of zebrafish may affect the effect of distress on the cortisol levels in whole-body homogenates. This indicates that stress in the feed reward group may have been experienced differently by the fish than in the other groups which may be one potential reason why comparison of cortisol levels in such an experiment is complicated.

  1. The manuscript examines the expression of numerous genes across different brain regions. It would be highly beneficial to provide a concise, conclusive summary of key patterns following each experimental result section. Merely listing the upregulation or downregulation of individual genes suffices to demonstrate transcriptional differences, but the interconnections and relationships among these genes across various stimulus treatments remain unclear. This lack of integrative analysis gives the manuscript a fragmented and unfocused feel. We strongly recommend rationalizing the underlying logic and research framework, and reorganizing the manuscript's structure and writing to enhance coherence and focus.

Answer: Thank you, we have re-organized the results section, now showing the genes for each pathway, not for each brain part. And for learning more from the data set, PCA, as unsupervised learning methods, have been performed to have a better view on the patterns of the gene expressions.

  1. Appendix A section should be revised to a standard three-line table.

Answer: the formatting of the tables has been revised.

Reviewer 4 Report

Comments and Suggestions for Authors

Authors should revise the abstract to more clearly describe the methodology and specify the genes that analysed. Also, highlight the aim of the study. 
Provide more accurate and relevant alternative keywords.
Lines 33-34, 45-48: Add references
Line 134: Revise the title 
Lines 148-150: Rewrite the sentence for clarity 
Line 158: Revise the title 
Lines 166-167, 170-171: These sentences are not suitable for this section. 

Line 451: Revise which makes it difficult 

Lines 453-459: Consider discussing briefly on why gene expression might offer a more nuanced view of stress response 

Lines 472-481: how Mr differs functionally from Gr in zebrafish.

Line 527: Revise the sentence for clarity 

Add a summary paragraph at the end of Section 4.3 to synthesize key findings across gene findings 

In conclusion, suggest future directions of research

Comments on the Quality of English Language

The manuscript is generally well-written and the scientific content is clear. However, the discussion section would benefit from improved clarity and flow  by incorporating smoothing transitions between sections. Additionally, some sentences require rephrasing for grammatical accuracy such as line 527. In conclusion, genes expression should corrected to gene expression and it would be helpful to specify which gene expressions are being referred to.

Author Response

Authors should revise the abstract to more clearly describe the methodology and specify the genes that analysed. Also, highlight the aim of the study. 

Answer:Thank you for this comment. The abstract has been revised accordingly. All changes in the manuscript are highlighted in yellow.

Provide more accurate and relevant alternative keywords.

Answer:the keywords have been changed

Lines 33-34, 45-48: Add references

Answer: This has been done.

Line 134: Revise the title 

Answer: This has been done.

Lines 148-150: Rewrite the sentence for clarity 

Answer: This has been done.

Line 158: Revise the title 

Answer: This has been done.

Lines 166-167, 170-171: These sentences are not suitable for this section. 

Answer: We don`t fully agree here. The sentence about the significance of the results is necessary, because some readers expect that all results (also non-significant ones) are shown. But this is not possible, because so many genes have been investigated. The second sentence has been changed.

Line 451: Revise which makes it difficult 

Answer: ok. This has been done.

Lines 453-459: Consider discussing briefly on why gene expression might offer a more nuanced view of stress response 

Answer: okay, some sentences have been added here.

Lines 472-481: how Mr differs functionally from Gr in zebrafish.

Answer: The different roles of Gr and Mr has been explained.

Line 527: Revise the sentence for clarity 

Answer: This has been done.

Add a summary paragraph at the end of Section 4.3 to synthesize key findings across gene findings 

Answer: Thank you. A new paragraph has been added.

In conclusion, suggest future directions of research

Answer: Thank you. We have added future directions.

Round 2

Reviewer 2 Report

Comments and Suggestions for Authors

I have no comments to the manuscript of this version.

Reviewer 3 Report

Comments and Suggestions for Authors
  1. Regarding the second question, the author replied that the experiment was repeated twice. For the content being studied in this research, the sample size is too small and thus has limitations. Moreover, the specific sampling method was not explained, and the statistical analysis used only two biological replicates to process the data, which is not rigorous enough.
  2. The sixth question did not receive any specific information about the changes made, nor did it receive an explanation on how the data was reorganized logically. Therefore, this question was not answered at all.